# The Development of Human Ex Vivo Models of Inflammatory Skin Conditions

**DOI:** 10.3390/ijms242417255

**Published:** 2023-12-08

**Authors:** Eddy Hsi Chun Wang, Rebecca Barresi-Thornton, Li-Chi Chen, Maryanne Makredes Senna, I-Chien Liao, Ying Chen, Qian Zheng, Charbel Bouez

**Affiliations:** 1L’Oreal Research and Innovation, Clark, NJ 07066, USA; 2Harvard Medical School, Boston & Beth Israel Lahey Health, Burlington, MA 01805, USA

**Keywords:** skin inflammation, ex vivo model, skin barrier

## Abstract

Traditional research in inflammatory dermatoses has relied on animal models and reconstructed human epidermis to study these conditions. However, these models are limited in replicating the complexity of real human skin and reproducing the intricate pathological changes in skin barrier components and lipid profiles. To address this gap, we developed experimental models that mimic various human inflammatory skin phenotypes. Human ex vivo skins were stimulated with various triggers, creating models for inflammation-induced angiogenesis, irritation response, and chronic T-cell activation. We assessed the alterations in skin morphology, cellular infiltrates, cytokine production, and epidermal lipidomic profiles. In the pro-angiogenesis model, we observed increased mast cell degranulation and elevated levels of angiogenic growth factors. Both the irritant and chronic inflammation models exhibited severe epidermal disruption, along with macrophage infiltration, leukocyte exocytosis, and heightened cytokine levels. Lipidomic analysis revealed minor changes in the pro-angiogenesis model, whereas the chronic inflammation and irritant models exhibited significant decreases in barrier essential ceramide subclasses and a shift toward shorter acyl chain lengths (<C18), indicating skin barrier instability. Additionally, the irritant and chronic inflammation models are responsive to immunosuppressants. These models hold promise for advancing scientific understanding and the development of therapeutic and skincare solutions for individuals afflicted by compromised skin conditions.

## 1. Introduction

Inflammatory skin diseases (ISDs) are characterized by the activation of innate and adaptive immune responses, accompanied by the secretion of pro-inflammatory cytokines and other soluble factors in response to various inflammatory triggers, both internal and external in origin. These conditions encompass a diverse spectrum, and it is estimated that up to 25% of the adult population is afflicted by at least one ISD [1].

Among the most prevalent ISDs are specific forms of eczema/atopic dermatitis (2–10%) [2,3], rosacea (~10%) [4,5], and chronic conditions like psoriasis (1–3%) [1,6,7]. It is noteworthy that variables such as age, gender, environmental factors, and ethnic background exert a substantial influence on the prevalence and predisposition to various ISDs [1].

In the realm of ISD research, various methodologies are commonly employed, each with its unique advantages and disadvantages. Animal and transgenic models, with a particular focus on rodent models, have proven invaluable for the investigation of psoriasis and atopic dermatitis. These models serve as pre-clinical models for comprehending disease mechanisms and facilitating drug discovery [8,9,10]. They allow for extended monitoring of disease progression within an in vivo context, offering insights into the biological intricacies of each condition under controlled conditions. Nevertheless, it is important to note that inherent biological disparities exist between mouse and human physiology, which may result in differential responses in transgenic models. Additionally, the presence of fur and the thinner nature of mouse skin compared to human skin can potentially influence the penetration of test compounds.

An alternative to animal models is the use of reconstructed human epidermis (RHE) models, which offer a fully in vitro approach to generating epidermal tissue closely mirroring the five distinct layers of human skin [11]. Variants of this model incorporate different cell types to emulate various aspects of skin physiology, including pigmentation and disease conditions [12,13,14,15]. Recent iterations of RHE models have incorporated additional cell types, such as fibroblasts, immune cells, or cytokines, thereby advancing progress toward the creation of full-thickness skin tissues in laboratory settings [14,16].

An alternative approach in modeling ISDs involves the utilization of human skin biopsies in skin explant organ cultures. This method has proven instrumental in the study of burn wound healing, fungal infections, and inflammatory responses [17,18,19]. Human ex vivo skin organ culture offers a paramount advantage in that it faithfully replicates the entire human-specific skin architecture, including the epidermis, dermis, skin-resident immune cells, vasculature, and other skin appendages. Consequently, this approach yields outcomes that are inherently very biologically relevant.

A pivotal strength of ex vivo skin culture is its ability to capture donor-specific attributes, including age and ethnic background, potentially offering a more personalized perspective on responses. Nonetheless, it is important to acknowledge that this donor-specificity also presents a significant limitation, as the level of variation between donors can be substantial, and the viability of skin explants in culture is comparatively short [18,20,21]. Despite these constraints, the ex vivo skin model remains a valuable tool for gaining in-depth insights into human skin biology and its responses to diverse interventions, and using them in conjunction with the RHE models will be a powerful approach to screening and validating experimental treatments. 

In this study, we present three distinct methodologies for inducing inflammatory responses in healthy human abdominoplasty skin biopsies. The primary objective of our research is to employ stimulants involved in the etiopathogenesis of common forms of ISDs to establish models capable of replicating the characteristic phenotypes associated with these conditions. Notably, we emphasize the perturbation of epidermal barrier integrity, building upon insights garnered from our prior investigations [22,23]. Our developed models offer the advantage of exhibiting a relatively rapid response to the applied stimulants, underscoring their potential as a robust platform for intervention studies. We firmly believe that these models hold significant promise in advancing scientific understanding and discovery pertaining to therapeutic options and skincare regimens for individuals afflicted by inflammatory skin diseases characterized by compromised epidermal barrier function.

## 2. Results

### 2.1. Inflammatory Triggers Resulted in Compromised Epidermal Barrier

Control explants, which received no treatment, exhibited an intact epidermis with minimal indications of spongiosis or cellular death by the end of the 7-day period. Application of the AG trigger (LL37) resulted in a slight degree of epidermal damage, with negligible disparity in overall skin morphology observed between low and high doses (Figure 1). However, when assessing the integrity of the cornified cell envelope in the stratum granulosum, a notable reduction and discontinuation in loricrin expression were observed in explants subjected to higher doses of the AG trigger, as compared to the control (Figure 2). This observation aligns with prior findings in patients diagnosed with papulopustular rosacea (PPR) [24], a condition often characterized by facial erythema and telangiectasia (spider vein) [25]. LL37 is an antimicrobial peptide with pleiotropic effects; it can modulate inflammation as well as vasculogenesis and plays a significant role in the development of rosacea [25,25].

Intriguingly, no discernible decrease in filaggrin expression was noted in the IHC. IHC revealed a significant increase in filaggrin protein expression within the epidermis when contrasted with the non-treated control (Figure 2). Comparable observations where the disparity in FLG expression within PPR cases was insignificant or even slightly elevated, with only marginal reductions in animal models, have been documented previously [24,26]. This suggests that the AG trigger employed did not uniformly disrupt skin barrier proteins but rather targeted specific pathways that bear a resemblance to those associated with rosacea in humans and reproduced a similar phenotype.

Examination of skin explants treated with the IR trigger (DNCB + IL4) unveiled a dose-dependent deterioration of the epidermis, accompanied by an increase in cellular density (Figure 1). Notably, both loricrin and filaggrin expression within the treated skin exhibited a dose-dependent reduction, with the highest trigger concentration yielding the most consistent results (Figure 2). DNCB is commonly used as a contact sensitizer to evoke allergic contact dermatitis or eczema in animal models and human clinical studies [27]. Eczema encompasses a multitude of subtypes, and atopic dermatitis is the most prevalent form. Patients typically manifest symptoms such as erythema, edema, and scaliness (lichenification), often leading to discomfort and itchiness, exacerbated by reduced trans-epidermal water loss [28].

Atopic dermatitis (AD) is characterized histopathologically by spongiosis and perivascular infiltration of T cells (especially Th2 response) and macrophages [28,29,30]. The normal expression of skin barrier proteins, including loricrin and filaggrin, is profoundly affected in AD, with genetic defects in the latter constituting a known risk factor for AD [28,31]. In the skin explants treated with a higher dose of the IR trigger, severe spongiosis was observed, alongside reduced expression of loricrin and disrupted expression of filaggrin (discontinuous expression along the epidermis, as compared to the control) (Figure 2). In our model, the addition of DNCB and polarization of Th2 via IL4 may create a contact dermatitis or AD-like state that irritates dermal and epidermal cells, which leads to a pro-inflammatory environment.

Morphologically, the CS-treated (PMA and Ionomycin) skin explants exhibited a phenotype closely resembling that of the IR-treated skin. A dose-dependent development of spongiosis and increased cellular density was evident when compared to the control (Figure 1). Given that the CS-treated skin was designed to mimic lymphocytic-mediated chronic inflammation via PMA/ionomycin stimulation, the extent of epidermal deterioration was notably severe, with expression levels of both loricrin and filaggrin displaying heightened disruption (Figure 2). Some of these findings closely paralleled observations in psoriasis cases, marked by decreased loricrin and filaggrin protein expression [32]. However, in this model, the cells were not polarized to Th17 to recapitulate the pathogenesis of psoriasis. This model represents a T-cell-mediated severely inflamed skin.

In summary, we have demonstrated three distinct methodologies that elicit varying degrees of compromised skin integrity by targeting specific biological pathways. Our data substantiate the feasibility of generating common manifestations of compromised skin, characterized by decreased expression of skin barrier proteins and heightened inflammatory infiltrates, through the application of inflammatory triggers.

### 2.2. The Inflammatory Phenotype Is a Result of Different Immune Cell Populations

Next, we investigated the presence of immune cells found in each of the models that were treated with the highest level of inflammatory triggers. We performed immunohistochemistry staining targeting CD45, CD68, and CD3-positive cells, as well as Toluidine Blue (TB) staining to assess mast cell degranulation. 

In the control tissue, as anticipated, we found a low signal for all the markers, accompanied by a sparse population of CD45, CD68, and CD3-positive cells. Toluidine Blue staining likewise revealed minimal mast cell activity (Figure 3). 

We observed a heightened presence of CD45- and CD68-positive cells within the AG-treated skin samples. Intriguingly, these CD45- and CD68-positive cells were detected in both the epidermis and the papillary layer of the dermis (Figure 3), indicating a potential recruitment or differentiation of skin-resident macrophages and possibly Langerhans cells, possibly mediated by LL37 [33,34].

The activation of CD45- and CD68-positive cells in the AG model may trigger the release of numerous pro-inflammatory cytokines and chemokines, such as IL1, IL6, and TNFa. Furthermore, LL37 treatment of the skin may induce the endogenous production of LL37 by keratinocytes, thus establishing a positive feedback loop. In addition to the aforementioned findings, we also observed a significant increase in mast cell degranulation within the papillary dermis in the AG-treated skin, a phenomenon consistent with the rosacea phenotype. Previous research has shown that LL37 can induce mast cell activation and the development of rosacea-like features in murine models [35]. Mast cell degranulation contributes to skin inflammation and irritation, further substantiating its relevance in this context.

In the IR-treated skin samples, we observed a presentation similar to that in the AG-treated skin, albeit with a notably higher intensity of the signal for CD45- and CD68-positive cells, as well as pronounced Toluidine Blue staining (Figure 3). Of particular interest, we detected a limited number of CD3-positive cells within perivascular regions, suggesting the possibility of a minor exocytosis event originating from monocytes that had remained within the blood vessels. This small presence of CD3-positive cells implies that the inflammatory phenotype primarily arises from macrophages rather than lymphocytes. In contrast to the AG model, we did not identify CD45+ cells within the epidermis of the IR-treated skin, indicating the recruitment of a distinct population of macrophages in the IR skin. Furthermore, the frequency of mast cell degranulation was notably higher in the IR-treated skin, evidenced by dark granules surrounding dark blue or purple cells (Figure 3).

Eczema and atopic dermatitis (AD) encompass diverse manifestations, with a complex interplay of immune cells contributing to the pathogenesis of the disease. AD is traditionally considered a Th2-driven condition, but it can evolve into a chronic inflammatory disease involving multiple immune response axes [36]. In our model, our objective was to emulate the acute irritation response by employing DNCB and IL4 to achieve a higher semblance of eczema. DNCB serves as a contact sensitizer and has previously been used in animal models to induce an eczema-like or contact dermatitis-like state [37,38]. On the other hand, IL4 is capable of polarizing the response toward a Th2 state while suppressing Th1 [39,40].

We believe that our model triggered a response akin to contact dermatitis or the acute phase of AD, as evidenced by the heightened presence of macrophages (CD68+) and mast cells (TB+). Activation of resident lymphocytes (CD3+) may necessitate additional time or specific triggers to establish a fully Th2-cell-mediated ex vivo model.

In the CS-treated skin, we harnessed a well-defined and potent mechanism for the non-specific activation of lymphocytes in vitro [41,42]. As anticipated, we observed an elevation in the numbers of CD45+ and CD3+ cells, particularly in close proximity to blood vessels, when compared to the control group (Figure 3). Notably, we managed to minimize the activation of macrophages and mast cells, with only a modest amount of CD68+ and TB+ signals detected. Significantly, our findings demonstrated that a state of chronic inflammation could be induced solely by activating skin-resident lymphocytes. It is worth noting that, in comparison to biopsies obtained from individuals with inflammatory conditions such as psoriasis or atopic dermatitis, our model exhibited a relatively lower presence of CD45+ and CD3+ cells (although the difference from the control was apparent). This discrepancy can be attributed to the inherent limitations of the ex vivo model, which lacks the ability to recruit a greater number of monocytes from the circulation [43,44]. Furthermore, the optimal activation method for achieving this outcome remains ongoing. Nevertheless, the in situ activation of skin-resident lymphocytes proved sufficient to induce the secretion of proinflammatory cytokines, ultimately resulting in damage to the epidermis.

Our investigations unveiled a fundamental distinction in the underlying mechanisms leading to compromised skin barrier integrity between the CS model and the AG and IR models.

### 2.3. The Inflammatory Skin Models Are Characterized by Different Cytokine Profiles

To corroborate the observed inflammatory phenotype in each of our models, we collected culture media samples at the conclusion of the 7-day culture period and conducted an evaluation of various cytokines, chemokines, and growth factors in comparison to the control, utilizing multiplex ELISA.

Surprisingly, despite the heightened presence of macrophages and mast cells in the AG model, we did not detect any significant differences in the entire spectrum of evaluated inflammatory cytokines. The duration of culture may affect the level of cytokines secreted, and further optimization may be needed for the AG model. Instead, we found a noteworthy increase in several angiogenic factors within the AG model by day 7, such as EGF, FGF2, and angiopoietin-1 (Figure 4A). Our model may be triggering early stages of angiogenesis in the ex vivo skin, which led us to employ 3D imaging throughout the tissue to identify potential signs of vasodilation or angiogenesis (Figure 4B).

In the IR model, we detected an increase in IL1a, IL1b, and TNFα levels (Figure 5A). This observation closely aligns with DNCB-induced rodent AD models, which exhibit heightened levels of IL1b, TNFα, and IL6 [37,38]. Interestingly, the level of IFNγ was similar to that of the control, which could be attributed to the antagonistic or Th2-polarizing effects due to the addition of IL4 [45,46]. Contact dermatitis and atopic dermatitis are known for their complex cytokine profiles, which can vary across different disease forms and the duration of the condition. AD is characterized by a Th2 cytokine profile but transitions into a combination of Th1 and Th2 responses in more chronic stages.

In the CS model, PMA and ionomycin bypass the need for physiological engagement of T-cell receptors in T-cell activation [41]. In our study, we observed a significant elevation in several pro-inflammatory cytokines associated with T cell activation, as well as keratinocyte responses to stress, including IL1a, IL1b, TNFα, and IFNγ (Figure 5B). Furthermore, we noted a marked increase in cytokines indicative of an exacerbated inflammatory response, such as MIP1b (induced by IL1b) and ICAM-1 (associated with inflammatory leukocyte recruitment). These cytokines collectively characterize a state of chronically inflamed skin driven by the activation of lymphocytes. Notably, our findings from this model underscore the potential to induce lymphocyte-specific responses in ex vivo skin samples, even in the presence of limited resident immune cells.

### 2.4. Inhibition of Inflammatory Responses with Pharmacological Immunosuppressants

To assess the potential of ex vivo skin for anti-inflammatory agent evaluation, we co-administered hydrocortisone with inflammatory triggers (IR and CS) over a 7-day culture period. CS-treated skin, as expected, exhibited severe inflammation. However, co-treatment with hydrocortisone reduced cellularity and prevented skin barrier damage, confirmed by reduced CD45 signals (Figure 6A).

In contrast, dexamethasone treatment in the IR model showed no significant anti-inflammatory effects). Yet, combining IR with hydrocortisone decreased skin barrier damage and CD45 signals (Figure 6B).

These findings demonstrate the ability to modulate immune cell activity within 7 days, capturing inflammation’s consequences through immunohistochemistry. Steroid-based treatments, commonly used for inflammation symptoms, emphasize the model’s clinical translational potential.

### 2.5. Stratum Corneum Lipidomic Profile Revealed Significant Alteration of Skin Barrier Ceramides and Fatty Acid Groups

#### 2.5.1. Evaluation of the Epidermal Lipidome in Inflamed Skin Models

The epidermal skin barrier is a dynamic structure susceptible to the influence of exogenous and endogenous factors, such as inflammatory dermatoses. The interplay between inflammation, abnormal lipid composition, and barrier disruption in such conditions is widely recognized [47]. It is theorized that the presence of inflammatory cytokines interferes with the functionality of key enzymes associated with lipid synthesis, such as glucocerebrosidase, acid sphingomyelinase, and elongase of very long fatty acid-1,4. In turn, the composition of the epidermal lipidome is disturbed, which leads to barrier impairment and further exacerbates inflammation in the skin. In the inflammatory dermatoses models, the alteration of the epidermal lipidome was first evaluated by changes to the total analyte composition (total CER, FFAs, CHOL, CHOL-S) through lipidomic analysis. It was revealed that only the tissue subjected to the skin irritation stimuli (IR Model) induced a significant increase in total CER, total CHOL, and total FFAs (Figure 7). Clinical manifestations of AD typically demonstrate a decrease in total ceramides and cholesterol sulfate analytes; it is possible that the tissue in this model is increasing ceramide production following excision into biopsy punches as a reparative mechanism [48]. Decreased cholesterol sulfate is also typically lowered following desquamation to enable the shedding of corneocytes, which is a limitation of the ex vivo tissue used in the developed models [49]. Further analysis into the populations that make up these analytes was conducted.

#### 2.5.2. Alteration to the Ceramide Subclass Distribution

The composition of specific ceramide subclasses can be attributed to specific functions in the differentiation process and permeability barrier. For example, levels of Cer[NP] have been found to be elevated during differentiation and have been correlated with transepidermal water loss, while Cer[EOS] and Cer[EOH] contribute to the lamellar structure of the lipid multilayer [50,51,52]. Evaluation of the ceramide subclass distribution revealed changes to the population of ceramides that contribute to proper barrier formation and function. Figure 8 demonstrates both the IR and CS models display similar shifts to the ceramide subclass distribution. There was a significant decrease in esterified ceramides such as Cer[EOS] and Cer[EOH] and a significant decrease in Cer[NP], Cer[NH], and Cer[AH]. Interestingly, there was a significant increase in Cer[NS], which is typical in inflamed skin undergoing barrier disruption in both models. These shifts align with those observed clinically in both contact and atopic dermatitis conditions of the skin [52,53]. Lesional skin of patients with AD has also demonstrated elevated levels of sphingosine (such as Cer[NS] and Cer[AS]) and sphinganine ceramide bases [49]. Chronically inflamed tissue in the CS model demonstrated a significant reduction in Cer[AP]. The AG model only demonstrated a significant decrease in Cer[EOH], while the other barrier essential ceramides demonstrated no significant shift compared to untreated control tissue. Previous studies suggest that rather than changes to the epidermal ceramide subclass distribution, lipid transporters were significantly downregulated in conditions with inflammation-induced angiogenesis, such as rosacea [24]. Interestingly, the relative levels of ceramides in all three models vary from what is expected in native human skin, which has been consistently observed in other ex vivo tissue models. This can be attributed to the tissue culture period and conditions [54,55].

#### 2.5.3. Alteration to the Ceramide Chain Length Distribution

Recent studies have indicated the significance of ceramide chain length, specifically the carbon chain length of the acyl moiety, which is implicated in proper skin barrier function. Longer acyl chains are crucial in providing proper lipid lateral packing and barrier impermeability; meanwhile, shorter chains have been associated with promoting apoptosis, incomplete barrier maturation, and inflammation in the skin [56,57,58]. The population of acyl chain length in the three inflamed skin models was analyzed (Figure 9). It was revealed that both the IR and CS models demonstrated a significant reduction of very long chain acyl groups (C24–C29) and a significant increase in C16 acyl chain length. It has been previously reported that patients with inflamed skin conditions such as atopic dermatitis and psoriasis maintain higher populations of shorter-chain ceramides [49,52,53]. Meanwhile, the AG model demonstrated no significant change to acyl chain length. It is possible that the enzymes associated with the elongation process of the acyl chain group are more negatively impacted by the presence of cytokines and chemokines in the tissue in the IR and CS models. 

## 3. Discussion

The economic toll of inflammatory skin diseases (ISDs) on the US healthcare system exceeds $75 billion, encompassing medical expenses, preventive measures, and medication costs [59,60]. Beyond the financial aspect, ISDs impose a substantial psychosocial burden on patients, often leading to conditions such as depression [61,62]. Clinical presentations of ISDs vary from mild to severe manifestations, including erythema, dryness, itching, scaly plaques, and inflammation, frequently resulting in compromised skin barrier [63,64]. Importantly, the underlying mechanisms driving these manifestations differ across various ISD conditions [65]. Consequently, a comprehensive understanding of these distinct conditions through robust experimental models is imperative to provide optimal medical and aesthetic care for patients with ISDs. 

This study introduces three distinct methods for inducing inflammation in healthy human ex vivo skin. By employing specific triggers known to disrupt biological pathways implicated in the pathogenesis of various ISDs, we have developed models that hold significant potential for studying these conditions, facilitating the development of targeted therapies and skincare regimens. 

In the IR model, we induced skin distress using a contact sensitizer, DNCB, commonly utilized in the induction of atopic or contact dermatitis in murine models [37]. This model replicates epidermal morphology and an inflammatory cell signature resembling common forms of eczema, with a cytokine profile polarized by IL4. Furthermore, we demonstrated that alterations in the epidermal lipidomic profile align with previous research [49]. This model opens avenues for research focused on targeting specific biological pathways associated with eczema, as evidenced by our ability to inhibit inflammation using hydrocortisone.

The CS model is distinguished by lymphocyte-mediated inflammation via the use of PMA and ionomycin that nonspecifically activates resident immune cells to produce a phenotype resembling chronic inflammatory conditions like psoriasis. Notably, a similar model has been recently developed by Genoskin (Genoskin, Toulouse, France), which further polarizes the activation of T cells into Th17 cells to emulate psoriatic skin for research purposes. This method of activation also holds potential for expansion into other applications, such as scarring alopecia or alopecia areata, where substantial inflammatory infiltrates are commonly found around hair follicles.

Ex vivo models, in many instances, offer a better representation of human skin due to their comprehensive architectural and cell signaling pathways. However, they are subject to donor-to-donor variations and, notably, a disconnect from peripheral lymphoid organs that supply additional immune cells. The immune response in ex vivo skin primarily relies on resident immune cells, which may limit the breadth of responses compared to those anticipated from in vivo models. Another point of future development is to understand the timepoint dynamics for the inflammatory cytokine in the ex vivo inflammation models, as this work emphasized the morphological characterization of the tissue at 7 days post stimuli. The contribution effects of different sexes, ages, and ethnicities are also interesting questions for future direction. The current reported work is a combination of six different donors consisting of different donor demographics (two Caucasians, three Hispanic, and one African American donor(s)). Although in this study, we have demonstrated the robustness of the model being reproducible across different donor demographics, it will require significantly more donors in order to truly elucidate any potential contribution effects from age and donor demographics.

The ex vivo models developed in this study present an alternative to animal testing while preserving the intricate integration of immune cells with the rest of the skin, a facet often absent in RHE or in vitro models. As previously described, all experimental models have their strengths and limitations, and they do not fully recapitulate complex human diseases. Nonetheless, there is an evident demand for models that bridge the gap between in vitro or RHE and in vivo models. We believe these models hold promise for a broad array of applications, particularly in assessing the impact of various targeted anti-inflammatory agents on compromised skin barrier and pathogenic biological pathways.

## 4. Material and Methods

### 4.1. Reagents and Preparation of Inflammatory Stimulants

All reagents were purchased from Fisher Scientific (Waltham, MA, USA) unless otherwise indicated. Complete media without stimulants were composed of Dulbecco’s Modified Eagle’s Medium (DMEM) with 10% FBS and 1% penicillin-streptomycin. 

### 4.2. Inflammation-Induced Angiogenesis (AG) Model

LL-37 antimicrobial peptide (NC0659600) stock was reconstituted with distilled water and aliquoted; an aliquot was then added to the complete media to achieve a final concentration of 5 mg/mL (Low dose), 10 mg/mL (Medium dose), and 20 mg/mL (High dose). 

### 4.3. Skin Irritation (IR) Model

1-Chloro-2,4-dinitrobenzene (DNCB; AAA1377436) was dissolved in a small amount of DMSO and diluted with complete media to a final concentration of 5 mg/mL and 10 mg/mL. Recombinant Human IL4 (PHC0044) was reconstituted in PBS and aliquoted, then diluted with complete media to a final concentration of 20 ng/mL and 40 ng/mL. DNCB 5 mg/mL with rhIL4 20 ng/mL was denoted as “Low dose”, while DNCB 10 mg/mL with rhIL4 40 ng/mL was denoted as “High dose”.

### 4.4. Chronic Inflammation (CS) Model

Cell stimulation cocktail 500× (50-930-5) was ready-to-use. It has an optimized concentration of PMA and ionomycin. Cell stimulation cocktail was directly added to the complete media with a final concentration of 0.5× (Low dose), 1× (Medium dose), and 2× (High dose). 

Hydrocortisone (Sigma-Aldrich, Burlington, MA, USA; H0888-10G) was dissolved in a small amount of DMSO and then diluted in complete media to a final concentration of 500 mM.

Fresh stimulants were made each time while replenishing culture media.

### 4.5. Human Ex Vivo Tissue Processing 

Fresh human ex vivo skin tissues were obtained from donors who had provided informed consent following abdominoplasty procedures. The tissues were obtained through BioIVT (Westbury, NY, USA) following an approved WCG IRB protocol (IRB# 20180798) and shipped to a BSC level 2 research laboratory within 24 h after the procedure where the experiments took place. All experiments were performed in accordance with relevant guidelines and regulations. Donor information is provided in Appendix A. Ex vivo tissues were prepared by excising the subcutaneous fat followed by punch biopsies with disposable 12 mm Skin Biopsy Punch (Fisher Scientific, NC9253254) and placed in 24-well transwell plates with permeable polyester membrane inserts (Fisher Scientific, 07-200-161). At least three biopsies were used for each condition.

### 4.6. Tissue Treatment and Culture

Skin tissues were divided into groups of three and placed in the 24-well transwell plates, and different concentrations of inflammatory stimulants reconstituted in 650 uL of culture media were added to the well on the first day. Base media without stimulants served as the control group. Tissues were replenished with fresh media with the stimulant on day 3 and day 5. On day 7, supernatant from the culture was collected and aliquoted, and the tissues were bisected and fixed in 10% neutral buffered formalin (Fisher Scientific 22-126-347) for 3 h and then switched to 70% ethanol for storage until histology.

In the experiments combining the inflammatory stimulant and hydrocortisone treatment, the same protocol was used. Hydrocortisone was added and replenished together with the inflammatory stimulant on days 1, 3, and 5, and the tissues were collected on day 7.

### 4.7. Histology and Immunohistochemistry

Skin tissues were processed into formalin-fixed, paraffin-embedded (FFPE) block following a standard protocol (Histowiz, Inc., Long Island City, NY, USA) prior to histological and immunohistochemical (IHC) staining. Hematoxylin and eosin (H&E) staining (Figure 1), along with immunohistochemistry targeting loricrin (LOR) and filaggrin (FLG), were employed as analytical tools to elucidate the impact on the epidermal tissue structure (Figure 2) with standard protocols. Briefly, after fixation, 5 mm sections of tissue FFPE blocks were adhered to high-binding glass microscope slides. IHC was performed on a Bond Rx autostainer (Leica Biosystems, Buffalo Grove, IL, USA) with heat-induced antigen retrieval specific to each antibody. Bond Polymer Refine Red Detection (Leica Biosystems) was used according to the manufacturer’s protocol. All sections were then counterstained with a hematoxylin nuclear stain. After staining, sections were dehydrated and film coverslipped using a TissueTek-Prisma and Coverslipper (Sakura, Torrance, CA, USA). Whole slide scanning (40×) was performed on an Aperio AT2 (Leica Biosystems). The conditions used for each antibody are summarized in Appendix A. Toluidine Blue staining for mast cells was performed following manufacturer’s protocol (Sigma-Aldrich, Burlington MA, USA; 89640). Histology and IHC were performed on three tissues from each condition for each donor and with at least three donors.

### 4.8. Tissue Clearing for 3D Imaging

Paraformaldehyde-fixed samples were sent to LifeCanvas Technologies for processing. They were preserved using SHIELD reagents (LifeCanvas Technologies, Cambridge, MA, USA) using the manufacturer’s instructions [66]. Samples were delipidated using LifeCanvas Technologies Clear+ delipidation reagents. Following delipidation, samples were labeled with 10 µg Goat anti-CD31 (R&D Systems AF3628, Minneapolis, MN, USA) using eFLASH [67] technology which integrates stochastic electrotransport [68] and SWITCH [69] using a SmartBatch+. After immunolabeling, samples were incubated in 50% EasyIndex (RI = 1.52, LifeCanvas Technologies) overnight at 37 °C, followed by 1 d incubation in 100% EasyIndex for refractive index matching. After index matching, the samples were imaged using a SmartSPIM axially swept light sheet microscope using a 3.6× (0.2 NA) objective (LifeCanvas Technologies, Cambridge, MA USA).

### 4.9. Lipidomic of Epidermal Ceramides

Following culture, the epidermal layers of the ex vivo skin explants were mechanically separated from the dermal layers using the Thomas Stadie-Riggs Tissue Slicer. The dermis was discarded, and epidermal layers were preserved at −80 °C for lipidomic analysis. Lipidomic analysis was conducted by Metabolon (Morrisville, NC, USA). The epidermal lipids in the skin explants were extracted using organic solvent and analyzed on a Waters UPC^2^/Sciex QTrap 5500 mass spectrometer SFC-MS/MS system in MRM mode using characteristic parent-fragment mass transition for each analyte trace. Quantification of the individual lipid species is based on the peak area comparison of the lipid species and that of their corresponding surrogate standards. Concentrations were calculated in pmol/mg tissue for lipid species and classes.

### 4.10. Multiplex ELISA Assays

Tissue culture supernatant was collected at the end of the cultures to evaluate the secretion of several cytokines, chemokines, and growth factors. A 13-plex inflammatory cytokine and a 10-plex angiogenesis ProcartaPlex panel were used in our experiments (ThermoFisher, Waltham, MA, USA). Sample preparation was carried out following the manufacturer’s protocol and scanned using a Luminex MAGPIX Instrument System (ThermoFisher). The supernatant from each tissue was run in duplicate wells with at least three tissues from each condition for each donor and with at least three donors.

### 4.11. Statistical Analyses

Data were assessed using Student’s *t*-test, and two-way ANOVA with multiple comparison Holm–Sidak test. The differences between groups were compared using GraphPad Prism 9.0.1 (GraphPad Software Inc., La Jolla, CA, USA)

Lipidomic data were extracted using MATLAB R2020a and analyzed using a one-way analysis of variance (ANOVA) followed by the Tukey post hoc test. The statistical analysis was conducted using GraphPad Prism.

## Figures and Tables

**Figure 1 ijms-24-17255-f001:**
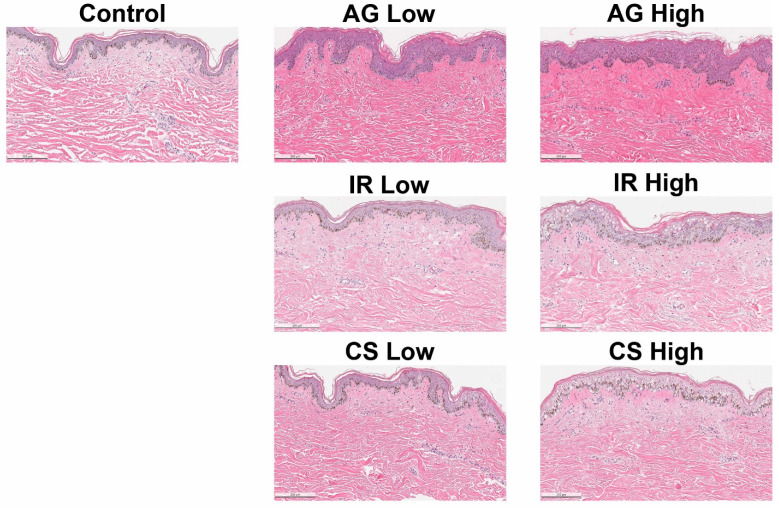
Morphology of inflammatory skin models. Representative H&E staining of human ex vivo skin biopsies treated with high and low concentrations of the stimulants (AG, IR, CS) compared to untreated control. Scale bar = 200 μm.

**Figure 2 ijms-24-17255-f002:**
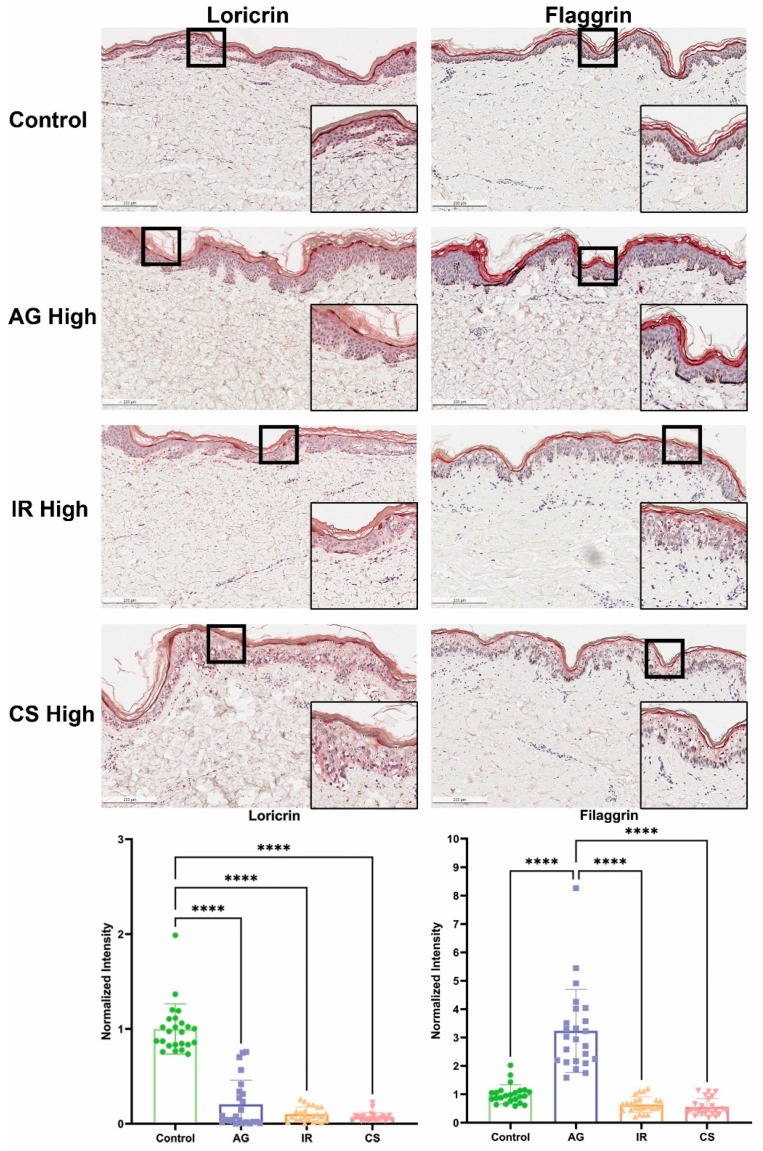
Assessment of epidermal barrier protein expression in the inflammatory skin models. Immunohistochemistry of loricrin and filaggrin signals in the epidermis of the models revealed an abnormal expression in all of the models at high concentrations. Quantification of the IHC signal revealed significant decrease of loricrin in all models but a significant increase of filaggrin in the AG model. A total of 24 measurements were made for each model (2 areas from each section, 3 sections from each donor, 4 donors). Scale bar = 200 μm, **** denotes *p*-value ≤ 0.0001.

**Figure 3 ijms-24-17255-f003:**
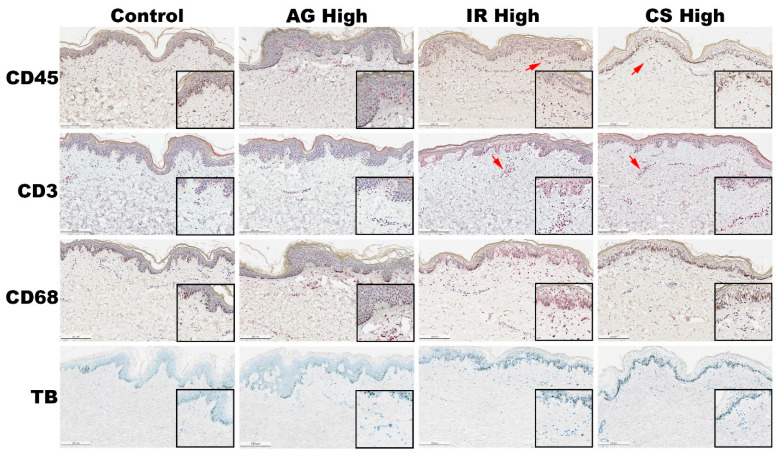
Assessment of different immune cell populations in the inflammatory skin models. AG model showed a higher signal of CD45 and CD68 in the epidermis; CD68 was also abundant in papillary dermis with higher presence of mast cells. IR model showed higher amount of CD68 and mast cells in the dermis but not in the epidermis. Similarly, most immune cells were found in the dermis of the CS model, but the level of CD45 and CD3 appeared to be higher than in other models and found around the blood vessels. Red arrows indicate positive signals near blood vessels. Scale bar = 200 μm.

**Figure 4 ijms-24-17255-f004:**
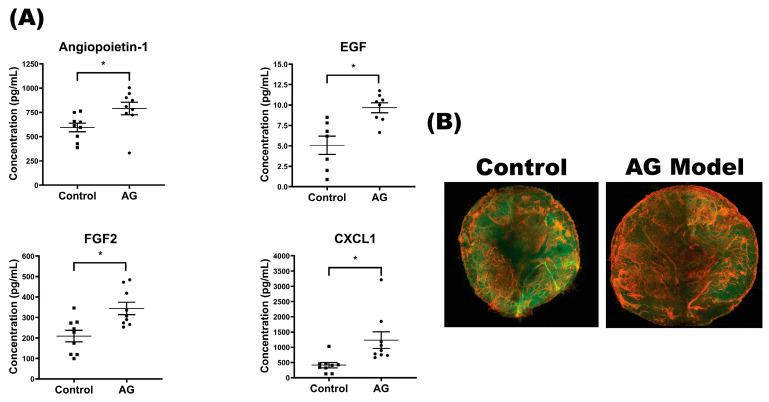
AG model showed a higher level of angiogenic growth factors and increased blood vessel branching. (**A**) Luminex revealed growth factors involved in angiogenesis and no changes in inflammatory cytokines. (**B**) The whole-mount staining of CD31, followed by 3D rendering of stacked images, showed higher blood vessel branching and larger diameter in the AG model. * denotes *p*-value ≤ 0.05.

**Figure 5 ijms-24-17255-f005:**
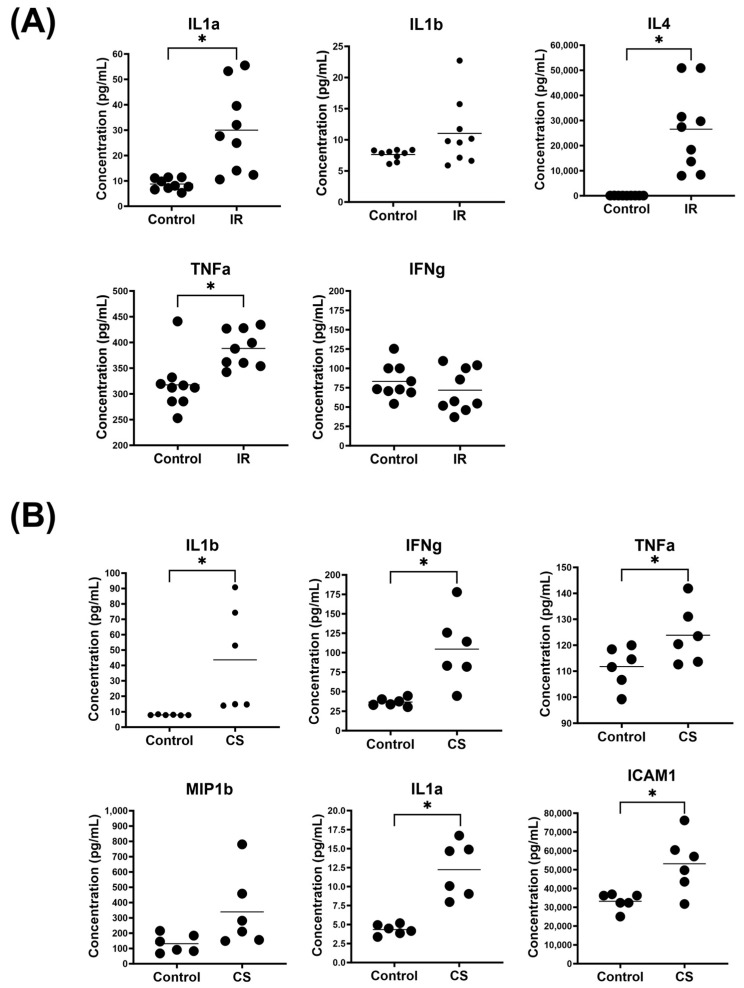
IR and CS models displayed a different panel of elevated cytokines. (**A**) IR model induced the secretion of inflammatory cytokines typically associated with distressed keratinocytes and a slightly suppressed IFNg secretion potentially through the modulation by IL4. (**B**) CS model induced a high amount of inflammatory cytokines involved in T-cell activation as well as chemokines involved in the trafficking of lymphocytes. * denotes *p*-value ≤ 0.05.

**Figure 6 ijms-24-17255-f006:**
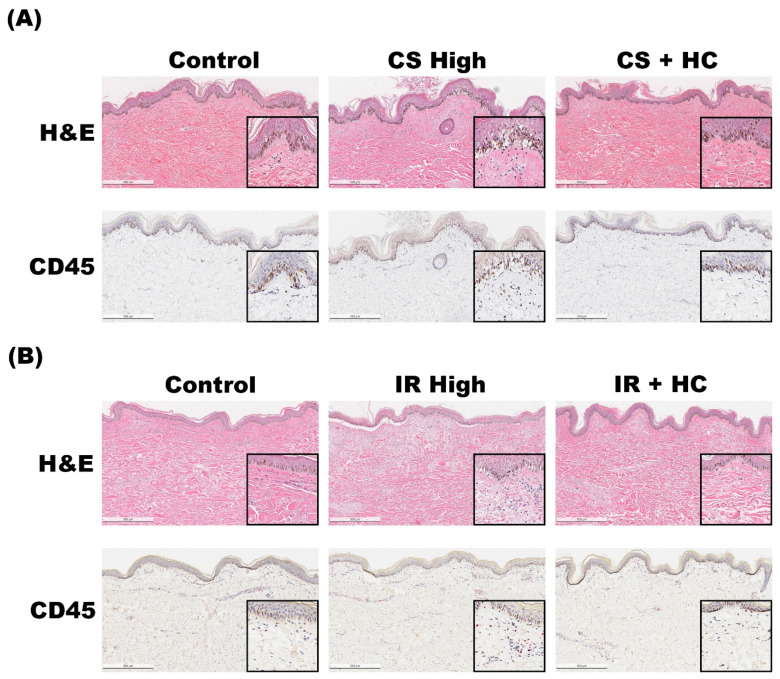
Inhibition of inflammatory phenotypes via hydrocortisone. Hydrocortisone effectively inhibited the deterioration of the epidermis induced by CS (**A**) and IR (**B**) stimulants. Further, the signal from CD45 cells was reduced to baseline level when hydrocortisone was co-treated with the triggers. Scale bar = 200 μm.

**Figure 7 ijms-24-17255-f007:**
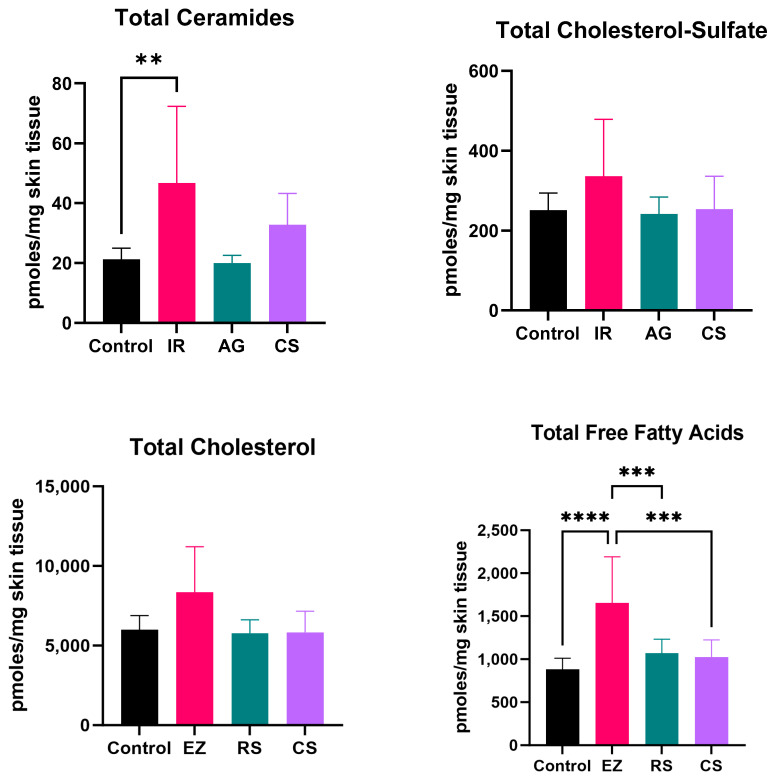
Epidermal lipidomic analysis of total analytes. IR model induced significant increase in total CER, total CHOL, and total FFAs compared to other models, suggesting a potential positive feedback of ceramide production to counter the IR stimulant. The AG and CS model did not show any significant changes. One-way ANOVA followed by Tukey post hoc analysis. ** *p* ≤ 0.01, *** *p* ≤ 0.001, **** *p* ≤ 0.0001.

**Figure 8 ijms-24-17255-f008:**
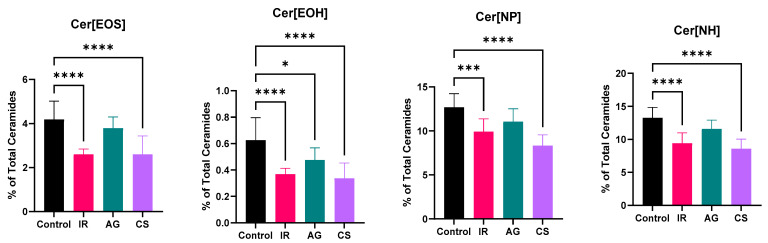
Analysis of the changes in ceramide subclasses. In all models, there was a shift towards lower Cer[EOS], [EOH], [NP], [NH], [AP], [AH] composition, and the CS model typically displayed a significant reduction that could lead to lamellar structure of the lipid multilayer disruption. The increase of Cer[NS] was found in all models, which has been reported in other inflammatory conditions. One-way ANOVA followed by Tukey post hoc analysis. * denotes *p*-value ≤ 0.05; ** *p* ≤ 0.01, *** *p* ≤ 0.001, **** *p* ≤ 0.0001.

**Figure 9 ijms-24-17255-f009:**
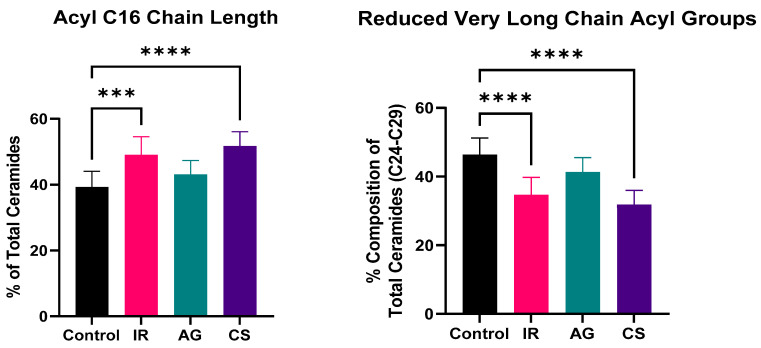
Altered acyl chain length distribution. In both the IR and the CS models, there was a significant trend toward increased shorter acyl chain length (C16), while a reduction of the very long acyl chain groups (C24–C29) was associated with impermeable barrier. The higher ratio of C16 to the longer chain groups implicates weakened lipid lateral packing, barrier impermeability, and incomplete barrier maturation. One-way ANOVA followed by Tukey post hoc analysis. *** *p* ≤ 0.001, **** *p* ≤ 0.0001.

## Data Availability

Data is contained within the article or Appendix A.

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
