# Peer review of "The Development of Human Ex Vivo Models of Inflammatory Skin Conditions"

_ijms, 2023, doi:10.3390/ijms242417255_

Round 1
Reviewer 1 Report
Comments and Suggestions for Authors
International Journal of Molecular Sciences
“The development of human ex vivo models of inflammatory skin conditions”
In the article by Eddy His Chun Wang et al, the authors introduce three distinct methods to induce inflammatory responses in healthy human skin ex vivo (human skin biopsies obtained after abdominoplasty). These developed models (Inflammation-induced angiogenesis-AG-model; Skin irritation-IR-model; Chronic inflammation-CS-model) represent an alternative to animal testing and also preserve the heterogeneity and integrity of the immune component, in vitro models as reconstructed human epidermis (RHE). The authors' primary objective was to use adequate stimulants involved in the etiopathogenesis of common forms of ISD to obtain models capable of replicating the characteristic phenotypes associated with these pathological conditions.
The originality of this article lies in the fact that the authors highlight the possibility to used human ex vivo skin organ culture that offers a paramount advantage in that it faithfully replicates the entire human-specific skin architecture, including the epidermis, dermis, skin-resident immune cells, vasculature, and other skin appendages. Consequently, this approach yields outcomes that are inherently very biologically relevant.
Furthermore, the authors, in the materials and methods section, correctly describe the procedures to obtain the three different inflammation models. The idea of studying other markers of inflammation could help improve the work with a view to considering other therapeutic targets. Furthermore, obtaining biopsies from other organs could contribute to the development of other human experimental models and verify/exclude any differences.
In addition, this article shed light on the possibility of personalized therapies to improve clinical management for individuals with inflammatory skin diseases characterized by impaired epidermal barrier function. The authors have a recognized experience in this filed.
The authors argue the data developed very clearly. The discussion and conclusions properly highlight the fundamental aspects of the research.
There are no references to add and those reported in the text are relevant to the topic.
Author Response
Thank you for reviewing our manuscript. We appreciate your time and supportive comments.
Reviewer 2 Report
Comments and Suggestions for Authors
The manuscript entitled ,,The development of human ex vivo models of inflammatory skin conditions’’ has been reviewed. In this paper, the authors deal with the important topic of inflammatory skin diseases (ISDs). The authors developed experimental models that mimic various human inflammatory skin phenotypes. The paper is in general well-written but it needs changes to be published
1. Section M&M –Human ex vivo tissue processing – Is there any information about donors? Gender and age? I understand that this is only a proposed ex vivo model that is an alternative to animal testing but the mentioned parameters may change the target result e.g. hormones secreted (gender) or the condition of the skin (age, aging skin) may affect the skin's response to administered stimulants.
2. Section M&M – AG model, IR model, CS model - in that section, doses were listed but the choice of doses was not explained. Whether the selection was based on the literature?
3. Assessment of epidermal barrier protein expression in the inflammatory skin model was limited only to loricrin and filaggrin signal detection in paraffin sections of the skin. What do RNA expression and protein levels look like in the analyzed skin models?
4. Whether the authors analyzed monocyte chemotactic protein (MCP1)?
MCP-1 induces the recruitment and activation of monocytes and macrophages during inflammation. MCP1 appears to be an important marker in the assessment of skin inflammation.
Author Response
Thank you for taking the time to review our manuscript and your valuable comments. Here are our responses to the questions you have.
- Yes, the donor demographics are included in Supplementary Table 2. All donors were female. 1 African American (54yo), 3 Hispanic (26yo, 60yo, 63yo), 2 Caucasian (45yo, 63yo). We understand that the age of donor may affect the response from the skin to the stimulants so we stayed consistent in the selection of donors to minimize other confounding factors. From our experiment, we did not see ethnicity and age being a variable for the models.
- The selection of the of the test doses is a combination of literature reference and prior experimentations (published and unpublished). For example, the use of LL37 in animal models (Ref 39). The use of DNCB and IL4 (Ref 41, 42), and PMA/Ionomycin (Ref 45 and also extrapolation of the recommended product concentration)
- The analysis of gene expression and protein synthesis is an excellent idea, and it is part of our future work to add to the models. We evaluated the changes in skin barrier integrity via lipidomic as our initial objective because of our prior work with ex vivo UV exposure model where the changes in different ceramide sub-classes and acyl chain length changes represent a robust assessment of barrier integrity (Ref 22, 23). There has been extensive research that studied the importance of ceramide and other stratum corneum lipids in skin barrier and during pathologic conditions, therefore we used this method to enable a broader and quantifiable evaluation at the overall changed (Ref 51-61).
- We did evaluate the release of MCP1 via multiplex ELISA. In both the IR and the CS models, MCP1 level showed an increasing trend in the inflamed models, however, they were not statistically significant. In the AG model, the release of MCP1 showed no difference compared to the control.
I hope these answer your questions, thank you again for your comments.